# Different Dosages of Progesterone in Luteal Phase Support Reflect Varying Endometrial microRNA Expression in Frozen Embryo Transfer Cycles

**DOI:** 10.3390/ijms25073670

**Published:** 2024-03-25

**Authors:** Wen-Jui Yang, Farn Lu, Cai-Yun Wang, Jun-Jie Hong, Tiffany Wang, Pok Eric Yang, Jack Yu-Jen Huang

**Affiliations:** 1Department of Infertility and Reproductive Medicine, Taiwan IVF Group Center, Hsinchu 30274, Taiwan; 2Department of Obstetrics and Gynaecology, Faculty of Medicine, The Chinese University of Hong Kong, Shatin, Hong Kong; 3Inti Labs, Hsinchu 30261, Taiwan; irenewang@intilabs.com (C.-Y.W.); tiffany@intilabs.com (T.W.);; 4Department of Obstetrics & Gynecology, Stanford University, Stanford, CA 94305, USA

**Keywords:** miRNAs, endometrial receptivity, progesterone, embryo transfer, in vitro fertilization

## Abstract

Despite serum progesterone being a widely accepted method for luteal phase support during embryo transfer cycles, debates persist regarding the optimal strategy for guiding clinical decisions on progesterone dosages to maximize reproductive outcomes. This retrospective study explored the utility of microRNA (miRNA) biomarkers in guiding personalized progesterone dosage adjustments for frozen embryo transfer (FET) cycles in 22 in vitro fertilization (IVF) patients undergoing hormone replacement therapy. Utilizing MIRA, an miRNA-based endometrial receptivity test, we analyzed patients’ miRNA expression profiles before and after progesterone dosage adjustments to determine suitable dosages and assess endometrial status. Despite patients receiving identical progesterone dosages, variations in miRNA profiles were observed in the initial cycle, and all patients presented a displaced window of implantation. Following dosage adjustments based on their miRNA profiles, 91% of patients successfully transitioned their endometrium towards the receptive stages. However, two patients continued to exhibit persistent displaced receptivity despite the adjustments. Given the evident variation in endometrial status and serum progesterone levels among individuals, analyzing miRNA expression profiles may address the challenge of inter-personal variation in serum progesterone levels, to deliver more personalized dosage adjustments and facilitate personalized luteal phase support in IVF.

## 1. Introduction

Progesterone (P4) plays a vital role in embryo transfer and sustaining pregnancy over the IVF process. Out-of-range serum progesterone in the late follicular phase of in vitro fertilization (IVF) cycles could detrimentally affect pregnancy outcomes, as it links directly to endometrial receptivity [1,2,3].

Despite the serum progesterone level being widely acknowledged as the conventional method for luteal phase support (LPS) during embryo transfer cycles, there are persisting debates about when to start, which is the best route, the dosage and duration of P4, and whether serum progesterone at the cut-off point of 10 ng/mL is associated directly with frozen embryo transfer (FET) outcomes [3,4,5,6]. In addition to the established findings that abnormal luteal progesterone levels diminish the likelihood of pregnancy outcome [1], research has also shown notable inter-personal variation in serum p4 levels that might influence reproductive outcomes, even when employing identical doses and routes of progesterone supplementation [3]. These results highlight the significant role of progesterone during embryo transfer and advocate the critical need to find more patient-specific endometrial biomarkers than the “one size fits all” serum progesterone level to deliver personalized dosage control over FET processes [3,5,7].

To date, more than 2500 microRNAs (miRNAs) have been discovered in the human genome. They play integral roles in diverse biological processes, including gametogenesis, embryogenesis, and the quality assessment of sperm, oocytes, and embryos, as well as diseases such as endometriosis, endometritis, and endometrial cancer [8,9,10,11]. Unlike messenger RNAs (mRNAs), miRNAs exhibit higher resilience to endogenous ribonuclease activity. Specific miRNAs like miR-30b, miR-181, and miR-223-3p govern endometrial cyclic changes, receptive state maturation, and crucial implantation markers. Dysregulated miRNAs, such as miR-21 and miR-22, may contribute to RIF by impacting early cell processes essential for implantation [8,9]. The MicroRNA-based Endometrial Receptivity Analysis (MIRA) platform integrates miRNA biomarkers relevant to fertility physiological processes and endometrial receptivity, offering a 95% predictive accuracy in identifying the displaced window of implantation as a contributing factor to recurrent implantation failure [9].

Recurrent implantation failure (RIF) remains a significant challenge in assisted reproductive techniques. Previous studies identified inadequate luteal phase support in the frozen embryo transfer (FET) cycle as one of the important causes of RIF and unsuccessful pregnancy outcomes [1,3,5,6]. Given this context, we wonder whether analyzing microRNA biomarkers from endometrium tissues could be applied to provide a more adequate luteal phase support in FET cycles. Therefore, the primary objective of this article is to investigate the alterations reflected in miRNAs due to different progesterone dosages. By analyzing miRNA profiles with MIRA, we aim to provide information that could contribute to the development of personalized dosage adjustment methods aside from monitoring serum progesterone levels and provide more individualized support during the luteal phase of FET cycles.

## 2. Results

From the MIRA, we can see that the miRNAs have a unique expression for the three endometrial stages (pre-receptive, receptive, post-receptive), as shown below [Figure 1]. This was validated by successful pregnancy results and was used as a baseline value in our subsequent heat map to analyze patients’ miRNA profiles.

The 22 patients included in this study all showed a displaced WOI (either pre- or post-receptive) in the initial mock cycle through MIRA’s analysis results [Table 1]. Specifically, two samples displayed a pre-receptive window, while the remaining 20 exhibited a post-receptive profile, implying that these patients did not receive their personalized optimal progesterone dosage and showed differing miRNA expression profiles from other patients while receiving the same progesterone dosage. 

Different endometrial stages for these patients could also be reflected in their miRNA expression profiles, with post-receptive patients exhibiting similar miRNA expression that differed from the pre-receptive patient’s miRNA expression [Figure 2]. 

Following subsequent dosage adjustments, the endometrial status in 20 out of the 22 cases reverted to receptive at 120 h, whereas No.3 and No.8 still showed a displaced window of implantation [Table 2]. This result implied that 91% of the patients had a correct personalized dosage adjustment. Among the unsuccessful cases, two patients transitioned from post-receptive to pre-receptive windows after the dosage adjustment.

As we were not certain whether the two patients who still presented a divergent window of implantation had no other individual physiological factors or progesterone sensitivity that might cause the persistence, we did not include them in our conclusive analysis in [Figure 3]. However, for the remaining 20 patients, discernible shifts in their miRNA expression profiles toward the stages of the WOI following dosage modification are evident in the heatmap [Figure 3]. The directional transition, progressing from the left to right side of the heatmap, suggests the efficacy of successful progesterone dosage adjustments (PDA) in achieving a more favorable alignment with the receptive stage.

## 3. Discussion 

Previously, optimal progesterone dosage during the luteal phase support was determined based on monitoring serum P4 levels. However, due to physiological differences in each individual, there is no consensus about a standard protocol for progesterone dosages and serum thresholds at 10 ng/mL that benefit reproduction outcomes [2,3,4,6]. 

In this study, patients who received the same progesterone dosage exhibited different miRNA expression profiles that correlated to different receptivity outcomes in the first mock cycle. This result implied that miRNA profiles could be utilized as unique biomarkers for reflecting individual endometrial status among patients even when they were administered the same dosage treatment. This personalized feature could potentially address the challenge of the inter-personal variation of serum progesterone levels seen from previous research [1,3,6] to guide decisions of progesterone dosage adjustments more accurately. 

While nearly 91% of patients exhibiting a displaced window of implantation successfully restored their endometrial stages to the standard 120 h post progesterone dosage adjustments, it is noted that two individuals showed endometrial stage alteration indicating different progesterone tolerance compared to the other patients, which could be adjusted accordingly to find the adequate progesterone dosage for embryo transfer. 

We investigated the changes in miRNA profiles among patients who received progesterone dosage adjustments, and they exhibited a shift in biomarker profiles as expected, which demonstrated a successful cohesion of their corresponding miRNA profiles toward the receptive stages based on MIRA. As miRNAs can influence and reflect the endometrium’s changes from the biological level, monitoring the optimal progesterone dosage in IVF patients could be tracked by utilizing miRNA biomarkers.

There is a growing amount of research emphasizing the potential role of miRNAs in personalized medicine within reproductive pathology and their evolving avenues for diagnostics and prognostics in fertility-related conditions such as gynecological cancer, polycystic ovary syndrome, and altered steroid hormone biology [12,13]. The result of this research could contribute to this evolving field by shedding light on a novel approach and insights into the utilization of miRNA in female health and assisted reproduction treatment by harnessing the power of the genetic encoding messenger to enhance patient-specific treatments. 

Since variation in the endometrium’s status and serum progesterone in each individual was evident both in our study and previous findings [1,3,6], solely relying on monitoring the serum P4 level might not provide accurate and comprehensive guidance for determining progesterone dosages. This study underscores the significant potential of utilizing miRNA biomarkers to monitor patients’ endometrium status and make informed decisions regarding the optimal progesterone dosage for each individual. miRNAs could potentially serve as more accurate biomarkers compared to serum progesterone alone, offering a promising alternative for predicting the adequate dosage of progesterone in the luteal phase support for frozen embryo transfer cycles. Such innovative findings could help to revolutionize traditional practices in determining the optimal progesterone dosage and addressing the clinical challenge of delivering personalized IVF treatment based on individual conditions. 

## 4. Materials and Methods

This study is a retrospective study that utilized MIRA, an miRNA-based endometrial receptivity test that analyzes close to 100 biomarkers, to look at the specific miRNA expression profile changes in IVF patients undergoing hormone replacement therapy (HRT) under a progesterone dosage adjustment (PDA) protocol for personalized embryo transfer (pET). Approval of this study was obtained from the Joint Institutional Review Board (JIRB, No.19-S-011-1) in 2019.

In the first mock cycle, 22 patients’ receptivity status and miRNA expression profiles were determined through MIRA by taking an endometrial tissue biopsy at 120 ± 4 h after progesterone administration vaginally (Utrogestan, 800 mg per day). Subsequent adjustments to the progesterone dosage were implemented if a pre- or post-receptive (indication of excessive or insufficient progesterone dosage) was identified. Among patients with a baseline pre-receptive window, the progesterone dosage was decreased from 800 mg to 600 mg vaginally in order to delay their WOI back to normal 120 h after progesterone administration. Similarly, for patients identified with a post-receptive window, their progesterone was increased from 800 mg to 1000 mg vaginally to speed up their WOI. In the second mock cycle, the cycle after dosage adjustment, a second endometrial tissue biopsy using MIRA was performed at P + 5 to assess whether the change in progesterone dosage altered the displaced WOI back to receptive at 120 h (indication of a proper progesterone dosage).

For data analysis, Analysis of Variance (ANOVA) was adopted in this study to understand the statistical changes in the miRNA profiles and the corresponding endometrial stages. 

## 5. Conclusions

From this research we can conclude that different progesterone dosages result in differences in both the endometrial stages and miRNA profiles across individuals undergoing frozen embryo transfer cycles. Given the intrinsic relationship between progesterone dosage and its corresponding plasma concentration, alterations in dosage inherently impact the other two biological factors. This potentially implies a novel approach that could deliver personalized progesterone dosage adjustments for optimal endometrial receptivity stages. Nevertheless, how the endometrial status and its associated miRNA profile could be influenced by dosage adjustment varies between individuals, highlighting the benefit of utilizing miRNA biomarkers to reflect the unique status of each individual. 

In conclusion, we found that different dosages of progesterone in luteal phase support affected the different expression of endometrial receptivity-related miRNA and could be adjusted to the optimal progesterone dosages. Personalized dosage adjustments (PDA) based on miRNA biomarker profiles might imply unique luteal phase support to identify the ideal dosage of progesterone for frozen embryo transfer cycles. Moreover, microRNAs (miRNAs) have the potential to monitor progesterone levels as a more reliable and accurate tool for guiding patient-specific progesterone dosage administration decisions. Further investigations are warranted to elucidate the precise relationship between progesterone dosages and corresponding variations in miRNA expression, providing a more comprehensive understanding for guiding dosage adjustments as a large proportion of this study presented a post-receptive window over the first mock cycle.

## Figures and Tables

**Figure 1 ijms-25-03670-f001:**
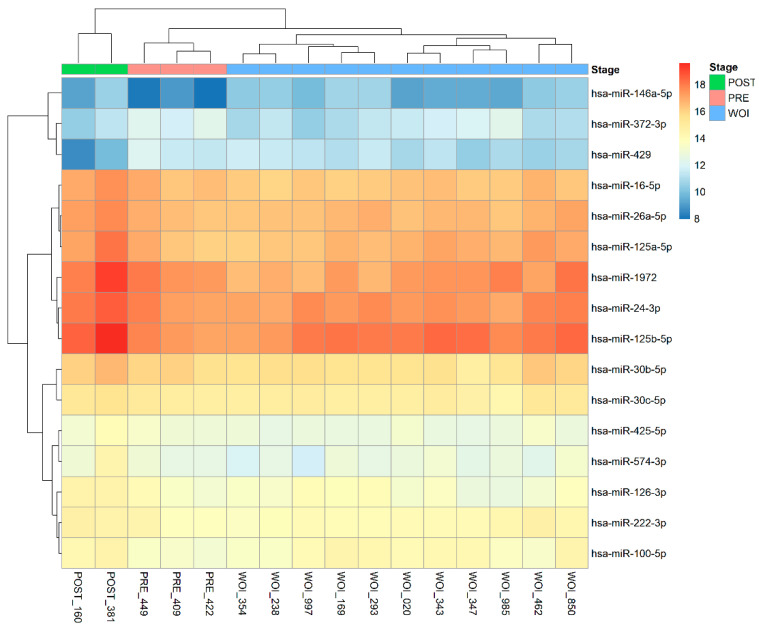
The heatmap of unique miRNAs expression profiles analyzed by MIRA for the three endometrial stages (pre-receptive, window of implantation, post-receptive).

**Figure 2 ijms-25-03670-f002:**
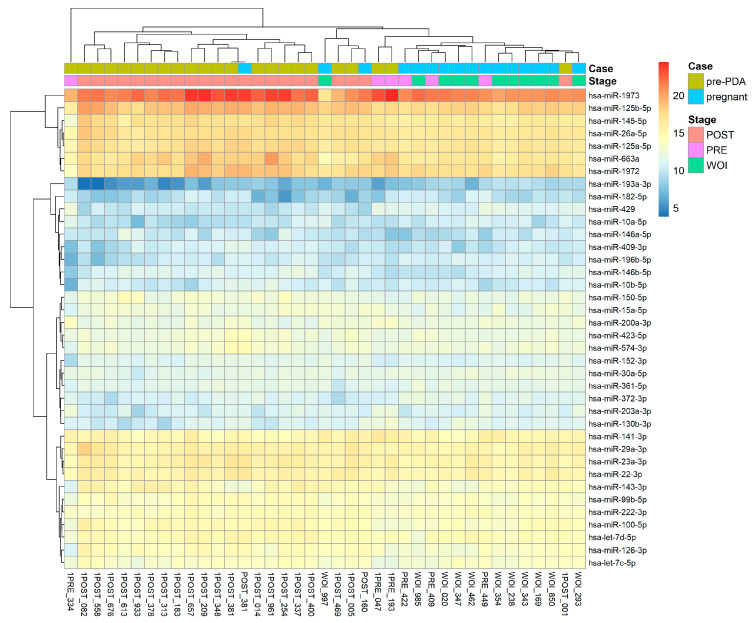
The heatmap of miRNAs expression profiles analyzed by MIRA for the 22 patients in the initial mock cycle, displaying a similar miRNA profile for post-receptive patients that is distinctive from pre-receptive patients.

**Figure 3 ijms-25-03670-f003:**
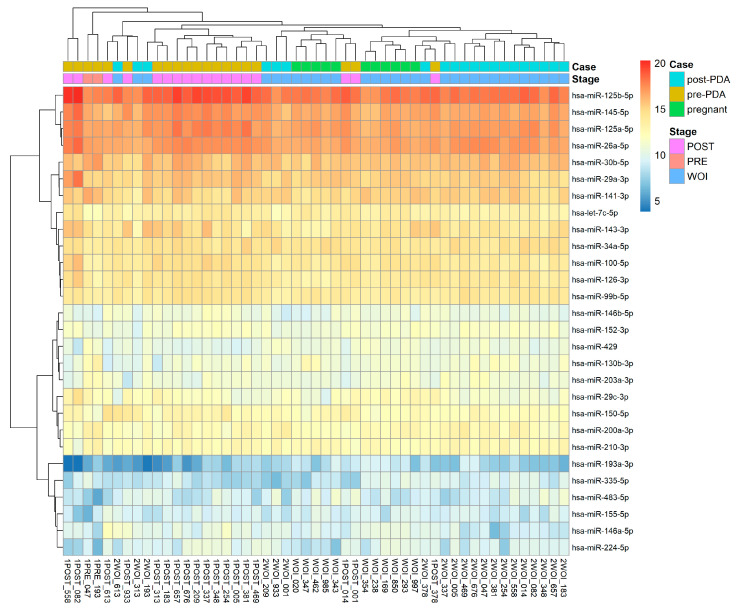
The heatmap of the miRNA expression profiles analyzed by MIRA for 22 patients from the initial mock cycle to post-dosage adjustments, showcasing a directional transition that suggests successful progesterone dosage adjustments in achieving a more favorable alignment with the receptive stage.

**Table 1 ijms-25-03670-t001:** MIRA’s analysis results of the individual endometrial receptivity stages for the 22 patients in the initial mock cycle.

Patient No.	Initial WOI
1	Post
2	Post
3	Post
4	Post
5	Post
6	Post
7	Post
8	Post
9	Post
10	Post
11	Post
12	Post
13	Post
14	Post
15	Post
16	Post
17	Post
18	Pre
19	Pre
20	Post
21	Post
22	Post

**Table 2 ijms-25-03670-t002:** MIRA’s analysis results of individual endometrial receptivity stages for the 22 patients from the initial mock cycle to post-dosage adjustments cycles, demonstrating 20 successful dosage adjustments cases with 2 exceptions.

Patient No.	Initial WOI	Post-Adjustments WOI
1	Post	WOI
2	Post	WOI
3	Post	Pre
4	Post	WOI
5	Post	WOI
6	Post	WOI
7	Post	WOI
8	Post	Pre
9	Post	WOI
10	Post	WOI
11	Post	WOI
12	Post	WOI
13	Post	WOI
14	Post	WOI
15	Post	WOI
16	Post	WOI
17	Post	WOI
18	Pre	WOI
19	Pre	WOI
20	Post	WOI
21	Post	WOI
22	Post	WOI

## Data Availability

Data are contained within the article.

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
