# Peer review of "Different Dosages of Progesterone in Luteal Phase Support Reflect Varying Endometrial microRNA Expression in Frozen Embryo Transfer Cycles"

_ijms, 2024, doi:10.3390/ijms25073670_

Round 1
Reviewer 1 Report
Comments and Suggestions for Authors
The article discovered that miRNA profiles and endometrial status can be influenced by different progesterone dosages and concluded that miRNA can be used as a biomarker for future personalized dosage adjustments. The article is well-written, but I think background information is missing. Minor revisions are suggested.
1. I would suggest adding the full name of IVF in the abstract to help readers better understand.
2. The figure legends are missing. Please consider adding those.
3. Though the authors described MIRA as a miRNA-based endometrial receptivity test, there is no background information regarding MIRA. I would suggest adding background information and the reason for choosing this test in the work to provide a better understanding for readers.
Author Response
Comment 1. I would suggest adding the full name of IVF in the abstract to help readers better understand.
Response 1. Thank you for highlighting this, we have added the full name of IVF in abstract for a clearer understanding for readers.
Comment 2. The figure legends are missing. Please consider adding those.
Response 2. Thank you for noticing the missing figure legends, relevant information of every figures/ table has been well supplemented.
Comment 3. Though the authors described MIRA as a miRNA-based endometrial receptivity test, there is no background information regarding MIRA. I would suggest adding background information and the reason for choosing this test in the work to provide a better understanding for readers.
Response 3. Thank you for pointing out the lack of description for the utilization for MIRA, we have added in more background information about MIRA and the reason for choosing this platform in the last two paragraph of introduction section for a clearer understanding.
Reviewer 2 Report
Comments and Suggestions for Authors
Dear Authors,
I was invited to review the article titled Different dosages of progesterone in luteal phase support reflect varying endometrial microRNA expression in frozen embryo transfer cycles, a rather meaningful and well-crafted analysis centered around miRNA expression profiles and the extent to which their assessment can address the challenge of interpersonal variation in serum progesterone levels, thus enabling a personalized approach to dosage adjustments and a patient-specific luteal phase support in medical-assisted procreation.
The article struck me as a well crafted and competently assembled piece of research with remarkable noteworthy strengths: novelty, relevance in ART research and even a potential appeal to a relatively broad scholarly readership; furthermore, as far as its stated objective is concerned, it is rather thorough and it is grounded in sound and well-outlined methodology, as far as I could establish; the table and figures are competently crafted as well, hence effective in conveying key data and findings.
Having said that, I would like to call attention to some shortcomings which should be addressed:
The article's objective should be enunciated more clearly, along with the key points the authors mean to illustrate and how such findings are meaningful to MAP pathways. Along the same lines, I strongly recommend more thorough contextualization and elaboration, within the boundaries set by "Communications" standards, on the significance of the study's findings, especially in terms of crafting new innovative prognostic/therapeutic avenues stemming from personalized/precision medicine, ncRnas, molecular classifications. Such highly innovative and still evolving avenues for diagnostics and prognostics have for instance already been harnessed as markers for the prognosis and treatment of diseases such as gynecological cancers affecting fertility prospects with a degree of success and remarkable potential development. It is also worth considering said innovations against the backdrop of the progress undertaken by novel technologies such as AI, machine-learning and data processing.
The Discussion is in fact the section which comes across as rather lacking compared to the level of insightfulness you achieved relative to the article's main objective.
By the same token, the "tailored support" mentioned by the authors should also be briefly discussed from the standpoint of policy-making and evidence-based guidelines. Therefore, when elaborating on the importance of tailored approaches, a higher degree of contextualization is warranted, or that remark will stay underdeveloped.
The following sources should be drawn upon and cited:
Zhang R, Wesevich V, Chen Z, Zhang D, Kallen AN. Emerging roles for noncoding RNAs in female sex steroids and reproductive disease. Mol Cell Endocrinol. 2020 Dec 1;518:110875.
Cavaliere AF, Perelli F, Zaami S, Piergentili R, Mattei A, Vizzielli G, Scambia G, Straface G, Restaino S, Signore F. Towards Personalized Medicine: Non-Coding RNAs and Endometrial Cancer. Healthcare (Basel). 2021 Jul 30;9(8):965. doi: 10.3390/healthcare9080965.
Glatstein I, Chavez-Badiola A, Curchoe CL. New frontiers in embryo
selection. J Assist Reprod Genet. 2023 Feb;40(2):223-234. doi:
10.1007/s10815-022-02708-5.
Overall, the manuscript is well-structured with an acceptable level of English and it can constitute a valuable contribution to a highly relevant ART research area and be attractive and valuable to a relatively broad readership. Further proofreading by a native speaker of English is still advisable.
Sincerely.
Comments on the Quality of English LanguageThe level of English is acceptably good. Further proofreading by a native speaker of English is still advisable.
Author Response
Comment 1: The article's objective should be enunciated more clearly, along with the key points the authors mean to illustrate and how such findings are meaningful to MAP pathways. Along the same lines, I strongly recommend more thorough contextualization and elaboration, within the boundaries set by "Communications" standards, on the significance of the study's findings, especially in terms of crafting new innovative prognostic/therapeutic avenues stemming from personalized/precision medicine, ncRnas, molecular classifications. Such highly innovative and still evolving avenues for diagnostics and prognostics have for instance already been harnessed as markers for the prognosis and treatment of diseases such as gynecological cancers affecting fertility prospects with a degree of success and remarkable potential development. It is also worth considering said innovations against the backdrop of the progress undertaken by novel technologies such as AI, machine-learning and data processing.
The Discussion is in fact the section which comes across as rather lacking compared to the level of insightfulness you achieved relative to the article's main objective.
By the same token, the "tailored support" mentioned by the authors should also be briefly discussed from the standpoint of policy-making and evidence-based guidelines. Therefore, when elaborating on the importance of tailored approaches, a higher degree of contextualization is warranted, or that remark will stay underdeveloped.
The following sources should be drawn upon and cited:
Zhang R, Wesevich V, Chen Z, Zhang D, Kallen AN. Emerging roles for noncoding RNAs in female sex steroids and reproductive disease. Mol Cell Endocrinol. 2020 Dec 1;518:110875.
Cavaliere AF, Perelli F, Zaami S, Piergentili R, Mattei A, Vizzielli G, Scambia G, Straface G, Restaino S, Signore F. Towards Personalized Medicine: Non-Coding RNAs and Endometrial Cancer. Healthcare (Basel). 2021 Jul 30;9(8):965. doi: 10.3390/healthcare9080965.
Glatstein I, Chavez-Badiola A, Curchoe CL. New frontiers in embryo selection. J Assist Reprod Genet. 2023 Feb;40(2):223-234. doi:10.1007/s10815-022-02708-5.
Response 1: Thank you very much for providing the insightful ideas and resources that could help us align more closely with the "Communications" standards, we have added in two citations in the last two paragraph of our discussion sections to more comprehensively elaborate how our innovative findings could provide insights in the reproduction area and complement the evolving findings within the genetic coding realm. We excluded the third references because we believe our study has yet touch upon the AI analysis therefore should focus on pure miRNA analysis for now. Additionally, as the current practice of progesterone have no consensus on a standard treatment/protocol, we have revised the ‘tailored support’ into ‘personalized support’ to emphasize how progesterone dosage decision could be based on individual conditions as more evidence are needed to establish a more solid standpoint of policy-making decision.
Comment 2: Further proofreading by a native speaker of English is still advisable.
Response 2: Thank you for your feedback about the quality of english. We have carefully reviewed and revised the English in accordance with your suggestions.
Round 2
Reviewer 2 Report
Comments and Suggestions for Authors
Dear Authors,
After going through the manuscript's revised version, I must say that you have made a praiseworthy effort to improve the depth and comprehensiveness of this article.
In light of its strengths (novelty, thoroughness, relevance), I am green-lighting it for approval.
Best regards.